# Visual Echoes: A Simple Unified Transformer for Audio-Visual Generation

## Abstract

In recent years, with the realistic generation results and a wide range of personalized applications, diffusion-based generative models gain huge attention in both visual and audio generation areas. Compared to the considerable advancements of text2image or text2audio generation, research in audio2visual or visual2audio generation has been relatively slow. The recent audio-visual generation methods usually resort to huge large language model or composable diffusion models. Instead of designing another giant model for audio-visual generation, in this paper we take a step back showing a simple and lightweight generative transformer, which is not fully investigated in multi-modal generation, can achieve excellent results on image2audio generation. The transformer operates in the discrete audio and visual Vector-Quantized GAN space, and is trained in the mask denoising manner. After training, the classifier-free guidance could be deployed off-the-shelf achieving better performance, without any extra training or modification. Since the transformer model is modality symmetrical, it could also be directly deployed for audio2image generation and co-generation. In the experiments, we show that our simple method surpasses recent image2audio generation methods. Generated audio samples could be found in this anonymous link.

## 1 Introduction

Generative AI has become the most creative and vibrant field in machine learning communities. There are a mass of image generative models (Rombach et al., 2022; Ramesh et al., 2021; 2022; Betker et al., 2023; Saharia et al., 2022) which could synthesize photorealistic images, and tremendous emerging works (Gafni et al., 2022; Hertz et al., 2022; Couairon et al., 2023; Brooks et al., 2023; Tumanyan et al., 2023) on personalized generative model which enable wide range of controllabilities for image generation. Most of these methods are based on diffusion model (Ho et al., 2020; Song et al., 2021), which is already proved to have better performance and scalability.

Along with the rapid progress in image generation area, there also emerges a bunch of works on other modalities, such as for video generation (Blattmann et al., 2023b; Guo et al., 2023; Singer et al., 2022; Blattmann et al., 2023a; Khachatryan et al., 2023) and for audio generation (Liu et al., 2023; Huang et al., 2023; Evans et al., 2024; Kreuk et al., 2022). However, most of these existing (conditional) generation works only accept text input for conditional generation. Multi-modality generation beyond visual and textual modalities, which is of high practical value for advancing creativity as well as machine perception, is relatively under-investigated. There have been few studies on audio2image generation (Sung-Bin et al., 2023; Qin et al., 2023), audio2video generation (Yariv et al., 2023), image2audio generation (Sheffer & Adi, 2023) and video2audio (Luo et al., 2023) generation.

To achieve more flexible generation task, there are rising efforts to achieve any-to-any generation within a single model. Those methods either utilize a well-pretrained large language model (LLM) (Sun et al., 2023b; Tang et al., 2023a; Sun et al., 2023a; Wu et al., 2023), or train multiple while composable single-modality diffusion models (Tang et al., 2023b), else or train a huge multi-modal transformer model from scratch (Lu et al., 2023). After training with paired multi-modal data (such as text, image, audio and or video, *etc.*), the model could implement pluralistic conditional generation, thus achieving any-to-any generation.

Although these any-to-any generation models cover a wide range of conditional generation tasks and have great generation results, they usually demand a huge amount of training data, along with a giant model which usually has

more than 1 billion parameters (especially for those using LLM), as well as many engineering techniques for large scale training.

In this paper, to fill in the blank in multi-modal generation beyond visual-textual modality, we investigate audio-visual generation, which is relatively under-investigated in the multi-modal research, specifically we focus on image2audio generation while the trained model could directly be deployed to audio2image generation and co-generation. Instead of designing a huge model, we choose to achieve it through a simple non-autoregressive masked generative transformer, as its training is intuitively simple with fewer parameters than diffusion model. Generative transformer is not fully explored in multi-modal generation (covering audio), and the current related work with generative transformer (Sheffer & Adi, 2023) has complex pipeline and unsatisfactory generation audio quality. In the proposed pipeline, the paired image/audio data will first be encoded into discrete tokens with the corresponding pretrained Vector-Quantized GAN (VQGAN) (Esser et al., 2021; Iashin & Rahtu, 2021). Then a transformer model will take as input the concatenated audio-visual tokens. The generative ability is achieved by mask denoising training with an iterative inference policy. After training, the model could achieve image2audio generation which is the main focus of the paper, as well as audio2image generation and co-generation. The method outperforms its competitors, which either use diffusion or autoregressive transformer.

We summarize our contributions as below:

- We propose an elegantly simple method, showing that a light generative transformer model is able to conduct image2audio generation effectively.

- The trained model could be directly applied to audio2image generation and co-generation, as well as image-guided audio inpainting/outpainting task.

- We show our simple method surpasses other image2audio generation methods. With classifier-free guide deployed off-the-shelf, the performance could be improved further.

## 2 Related Works

**Multi-Modal Generation** In recent years, there emerge enormous text2image generation methods, most of them(Rombach et al., 2022; Ramesh et al., 2021; 2022; Betker et al., 2023; Saharia et al., 2022) are based on diffusion models. Considering the success of image generation, recently the community is getting increasingly interested in text2video generation (Blattmann et al., 2023b; Guo et al., 2023; Singer et al., 2022; Blattmann et al., 2023a; Khachatryan et al., 2023), where the boundary has been pushed forward a lot in the past year. Beyond visual and language modalities, there are also some recent works on audio and visual modalities. For image2audio generation, Im2Wav (Sheffer & Adi, 2023) uses cascaded autoregressive transformers with CLIP visual features as condition, and DiffFoley (Luo et al., 2023) trains a latent diffusion model with a contrastive audio-video pretrained model to encode the video information. For audio2image generation, Sound2Scene first learns to align image and audio features and then the audio features will be used to generate image by a pretrained BigGAN (Brock et al., 2018). There are also some works (Sun et al., 2023b; Tang et al., 2023a; Sun et al., 2023a; Wu et al., 2023; Zhan et al., 2024) resort to a pretrained large language model (LLM) to process multiple modalities to achieve any-to-any generation, which may include image (RGB, depth, segmentation map), text, sound, speech and video. There are also a few works (Ruan et al., 2023; Tang et al., 2023b) using multiple diffusion models to achieve generation across modalities. In this paper, we show a simple generative transformer could also conduct audio-visual generation.

**Generative Transformer** There is also a research line to implement generation with generative transformer for image, video, audio or multi-modal generation (Chang et al., 2022; Li et al., 2023; Yu et al., 2023; Chang et al., 2023; Mizrahi et al., 2023; Kim et al., 2023; Yu et al., 2024; Lu et al., 2023). As MAGVIT-v2 (Yu et al., 2024) suggests, using masked transformer for visual generation have several advantages, such as compatibility with LLM and visual understanding benefit. Our method is a masked generative transformer, and it could conduct bidirectional audio-visual generation within a single model. The most related works to ours are 4M (Mizrahi et al., 2023) and Unified-IO2 (Lu et al., 2023). 4M extends the masked generative transformer MAGE (Li et al., 2023) pipeline to multiple modalities with several modifications in the transformer, while not including audio modality. Unified-IO2 (Lu et al., 2023) is an autoregressive encoder-decoder transformer trained with mixture of objectives. Unified-IO2 could deal with audio, but it can only generate audio of 4s while we can produce 10s audio, and the training pipeline is complex with several training tricks

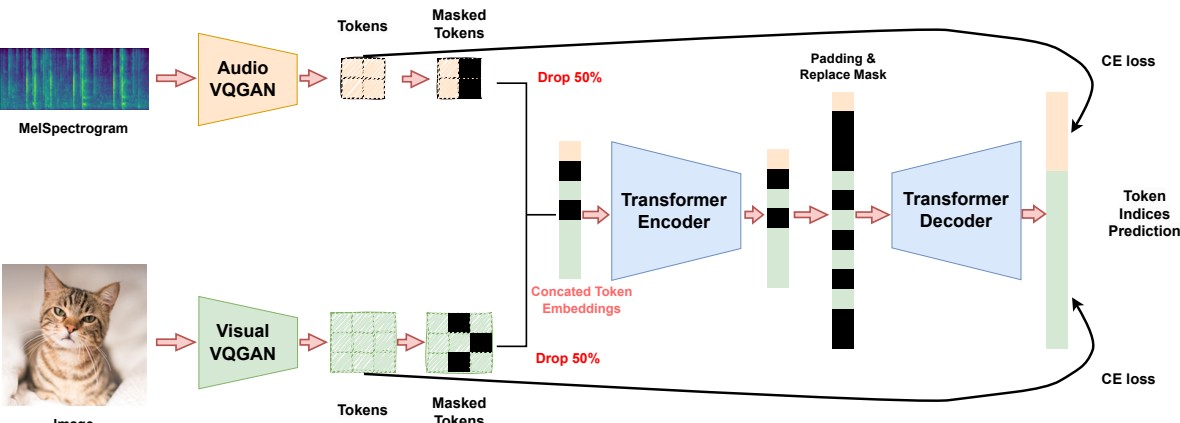

Figure 1: During training, we will randomly mask and drop 50% tokens, which may be either masked or not masked. We will pad the dropped positions (even if it is not masked originally) and masked positions with a learnable embedding before the transformer decoder.

or architecture modification. Compared to 4M and UnifiedIO-2, our non-autoregressive pipeline, which is a typical ViT (Dosovitskiy et al., 2020), is elegantly simple without extra module and overmuch architecture modification.

## 3 Method

Our method is based on a full-attention transformer, which operates in the discrete VQGAN space. In this section, we will first illustrate how we preprocess the audio/image data and the tokenization, then we will elaborate the transformer model and the training details, finally we will discuss how to do inference after training.

### 3.1 Data Preprocess

In this paper, we use the paired audio-image data from each 10-second video from VGGSound dataset (Chen et al., 2020). For image, we will sample 10 frames uniformly within a video, the resolution of the final visual frame $V$ sent to the model is $256 \times 256$. For audio, every 10 second waveform from the video will be transformed to mel spectrogram $A \in \mathcal{R}^{C_a \times T}$, where $C_a$ is the mel channels (frequency) and $T$ is the number of frame (time), we will use $A \in \mathcal{R}^{80 \times 848}$ in this paper. We sample paired image (random one out of 10 frames) and audio samples from the same video for model training.

### 3.2 Discrete Tokenization

In this paper, we do not put transformer on the raw image or mel spectrogram, instead we follow the masked generative transformer works (He et al., 2022; Li et al., 2023). We first encode image and audio to discrete tokens and transformer will operate in this latent discrete token space. The discrete tokenization through the pretrained VQGAN speeds up training compared to using raw input. The tokenizer we use are ImageNet pretrained VQGAN (Esser et al., 2021) for image where the codebook size is 1024 and codebook embedding dimension is 256, and AudioSet (Gemmeke et al., 2017) pretrained SepcVQGAN (Iashin & Rahtu, 2021) for audio where the codebook size and embedding dimension are the same as image VQGAN. Hereafter, we will refer to both image VQGAN and audio SpecVQGAN as VQGAN for simplicity. The encoder part of VQGAN will conduct the discrete tokenization resulting in discrete indices, and VQGAN decoder will decode the indices to image or audio mel spectrogram. Specifically, the image VQGAN will transform the image $V$ of resolution $256 \times 256$ to $16 \times 16$ discrete tokens, where each token is the index in the codebook. The audio SepcVQGAN will transform the audio mel spectrogram $A$ of size $80 \times 848$ to $5 \times 53$ discrete tokens. Those indices matrices will be further reshaped to 1D sequence $v \in \{1, ..., C\}^{L_v}$ and $a \in \{1, ..., C\}^{L_a}$, where $C$ is the corresponding codebook size in VQGAN, and note it is 1024 for both visual and audio VQGAN, and $L_v = 256$, $L_a = 265$ are the token length of image and audio. The output 1D sequence of the VQGAN encoders will be sent to the transformer model, and the output of the transformer will be decoded by the corresponding VQGANs to generate images and audio mel spectrograms. During training, all tokenizers are fixed.

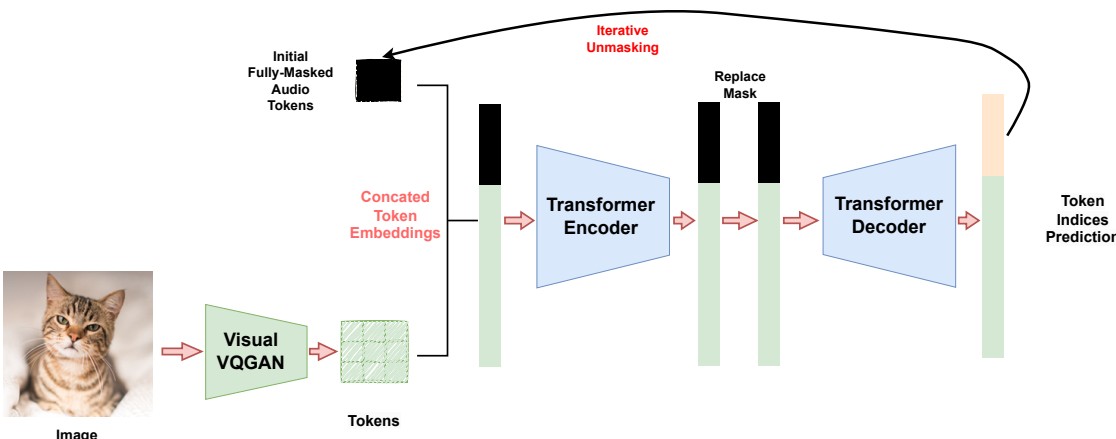

Figure 2: Conditional generation pipeline by iteratively unmasking during inference.

## 3.3 Audio-Visual Masked Generative Transformer

As mentioned before, the transformer will operate in the discrete VQGAN space. The raw image and audio mel spectrogram will first go through VQGAN encoder, leading to discrete tokens $a, v$. Then, these tokens are passed through an extra embedding layer (one for each modality), and converted into continuous token embeddings $f_a = E_a(a), f_v = E_v(v)$, where E denotes embedding layers, as we follow BERT (Devlin et al., 2018) style just like most generative transformer works (Esser et al., 2021; Chang et al., 2022; Li et al., 2023). The input to transformer are the directly concatenated audio and visual token embeddings $[f_a, f_v]$. Analogously, the output of the transformer layer, which is also continuous embedding, will be quantized resulting in token probability by computing dot product similarity between output embedding and weight of embedding layer. The transformer output has size of $p_a \in \mathcal{R}^{L_a \times C}$ and $p_v \in \mathcal{R}^{L_v \times C}$ for audio and image modalities, where $L$ is the token length (256 for image and 265 for audio) and $C$ is the size of VQGAN codebook (1024). Through this way, the discrete space is connected to the continuous embedding space.

**Transformer Architecture** We adopt the ViT architecture (Dosovitskiy et al., 2020) but without using $[cls]$ token, note here all attention layers are full attention. Similar to other masked transformer methods (He et al., 2022; Li et al., 2023; Mizrahi et al., 2023), we split the transformer model into 2 parts, encoder and decoder, to fit the masked modeling training paradigm. We will add learnable positional embedding and modality embedding (which could be regarded as bias), which are randomly initialized and learnable. They are added for each modality separately before the encoder and decoder. The architecture is similar to UniDiffuser (Bao et al., 2023) in some degree, while our pipeline is simpler and we use masked transformer instead of diffusion model.

**Mask Denoising Training** As mentioned before, the input to the transformer is the concatenated token embedding $[f_a, f_v]$, and the output of the transformer will be taken to compute similarity between the weight of embedding layer, the resulting similarity $[p_a, p_v]$ means the token probability, and they are corresponding to the original VQGAN encoder output.

To empower the transformer with generation ability, like other masked generative transformer methods (Chang et al., 2022; Li et al., 2023), we adopt mask denoising to train the transformer. Before concatenating and sending the token embedding into the transformer, we randomly mask a certain ratio for both audio and image independently, where the masked token is replaced with a special mask token from the embedding layer. Similar to MAGE (Li et al., 2023), the mask ratio is sampled from a truncated Gaussian, with range $[0.2, 1]$ and mean as $\mu$ (independently for audio and image), where $\mu$ is a hyperparameter. We will randomly drop 50% token embeddings for speeding up training and resulting in almost the same length of audio and image tokens. Note that during training, the tokens embedding may also contain the masked ones. Accordingly, we insert one extra mask embedding into the embedding layers of both audio and image.

As shown in Fig. 1, after masking and dropping, the token embedding, with positional embedding added, from two modalities will be concatenated and sent to transformer encoder. Before passing the transformer decoder. We will pad the audio and image token embedding with a modality-shared learnable mask embedding $m$, we will also replace

the remaining mask embedding coming from the encoder input with this learnable mask embedding. After adding positional embedding and modality-specific embedding for each modality, the resulting token embeddings will go through the transformer decoder. Following MAGE, the final output $[p_a, p_v]$ is the predicted token probability, which is the dot product between the transformer output and weight of the embedding layer (separately for each modality), followed by softmax operation. Here the weight of the embedding layer acts as a mapping layer without extra training. The training objective is the below cross-entropy loss on the predicted tokens which are masked, where the ground truth is the original indices output $a, v$ by the VQGAN encoder.

$$L = \mathbb{E}_{(A_i, V_i)}[M_a^i \mathcal{L}_{ce}(p_a^i, a_i) + M_v^i \mathcal{L}_{ce}(p_v^i, v_i)] \tag{1}$$

where the $A_i, V_i$ denotes *i-th* pair of audio and image data taken from the same video, and $p_a^i \in \mathcal{R}^{L_a \times C}$ denotes the audio token prediction from the transformer, and $a_i \in \{1, ..., C\}^{265}$ denotes the VQGAN encoder. The definition is the same for image modality. $M_a^i, M_v^i$ denote the binary mask used for masking the audio token, where 1 means mask and 0 means unmask, multiplying $M_a^i, M_v^i$ to the CE loss means we only consider loss on the masked token position. Minimizing this loss is to encourage the model to reconstruct the masked position with the information from the unmasked position of both audio and image modalities. The reason for using lower bound 0.2 instead of 0.5 in (Li et al., 2023) for the truncated Gaussian distribution, where the mask ratio is sampled, is to avoid the situation that both modalities are masked with quite high ratio, where the training may be unstable as there is little information from one modality to reconstruct the other modality.

**Relation to conditional diffusion model** Taking a closer look at a term of one modality in Eq. 1, $M_a^i \mathcal{L}_{ce}(p_a^i, a_i)$ measures the different between the prediction for masked token and groundtruth token, and $p_a^i$ could be rewritten as $p_a^i = p(a_i, v_i)$, which denotes the unmasked/denoised prediction from a masked (noised) input conditioned on the guided modality $v_i$, where the condition is directly treated an input. Therefore, it is similar to the current diffusion-based text2image/text2audio generation model, either continuous (Rombach et al., 2022; Liu et al., 2023) or discrete (Gu et al., 2022; Yang et al., 2023) ones. However, unlike those methods where the conditioning is unidirectional, the modality conditioning in our pipeline is symmetrical, which enables our method to conduct bidirectionally conditional generation within a unified model without extra training, additionally the model could also conduct image-audio co-generation.

Code 1: Iterative Unmasking

```python
def mask_by_random_topk(mask_len, probs, temperature=1.0):
    mask_len = mask_len.squeeze()
    confidence = torch.log(probs) + torch.Tensor(
        temperature *
        np.random.gumbel(size=probs.shape)).cuda(device=probs.device)
    sorted_confidence, _ = torch.sort(confidence, axis=-1)
    # Obtains cut off threshold given the mask lengths.
    cut_off = sorted_confidence[:, mask_len.long() - 1:mask_len.long()]
    # Masks tokens with lower confidence.
    masking = (confidence <= cut_off)
    return masking
```

## 3.4 Inference

Through training, the model will learn how to unmask/denoise the masked input for both image and audio, more importantly, the unmasking for one modality should consider the information from the other modality. This is achieved by the cross-modality attention in the self-attention layer inside transformer. Since the tokens going through the transformer are always the concatenated audio-visual tokens, the self-attention actually already computes the cross-modality attention. In this paper, we will focus on image2audio generation, while the model can also conduct audio2image generation and co-generation with the same inference policy. The output token indices from the transformer (based on the predicted token probability) will be sent back to corresponding VQGANs decoder to produce image and mel spectrogram. Additionally, mel spectrogram will be processed by a HifiGAN vocoder (Kong et al., 2020) to get waveform audio.

**Conditional Generation** As shown in Fig. 2, the conditional generation, either image2audio or audio2image generation, starts from fully masked tokens of the target modality, while the groundtruth token embedding from the guided modality will be used as condition, and they are directly concatenated to the tokens of target modality. We follow the iterative unmasking policy from (Chang et al., 2022; Li et al., 2023; Mizrahi et al., 2023), where in every iteration a certain ratio of the transformer predictions token will be updated to the input with remaining tokens kept masked, after $N$ steps the

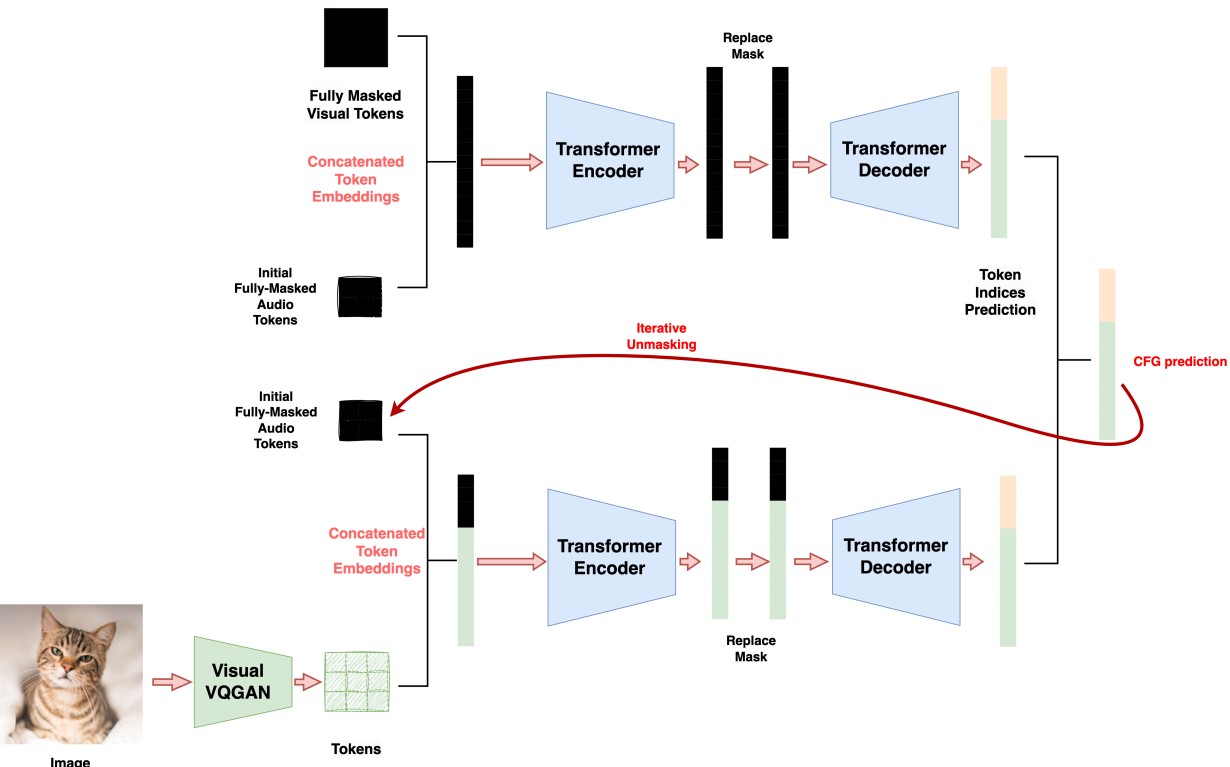

Figure 3: Classifier-free Guidance (CFG) on image2audio generation. There are actually 2 forward pass into the transformer, which are the different combination of audio and visual tokens.

full image or audio mel spectrogram are generated. Here the iterative unmasking policy has 2 hyperparameters, $T$ and $N$, where $T$ is the temperature to control the unmask speed.

We show the detail of the iterative unmasking in Code. 1. The indices predictions output by the transformer will not directly be used. Instead, it will be used as prior distribution of a categorical distribution, we will sample a new indices from this categorical distribution and use the new indices for iterative unmasking. We use the same unmask policy for image-audio co-generation, where the model generates image and audio token simultaneously in every iteration.

**Classifier-free Guidance** Classifier-free guidance (CFG) (Ho & Salimans, 2022) is proved to be a strong tool to achieve trade-off between generation fidelity and (conditional) alignment in diffusion-based conditional generation methods. In our paper, we also deploy CFG to improve the performance. However, unlike existing works that need to do extra unconditional training (typically remove the condition during training of a low ratio), CFG could be deployed to our trained model off-the-shelf, it is similar to UniDiffuser (Bao et al., 2023) which also uses concatenated embeddings as the input. We posit the reasons why CFG could directly work without extra training are 2-fold. First, the mask ratios (usually larger than 0.5) of two modalities are sampled independently, there is a chance that one modality is almost fully masked. Second, one necessary reason for extra CFG unconditional training in diffusion model is to learn denoising under the null condition (since most methods adopt text as condition) which is not seen during conditional training. While in our method, we have a learnable mask embedding which is put before the transformer decoder during training, which is used as unconditional input. In other words, the model already saw the unconditional input (the learnable mask embedding) during training. The off-the-shelf CFG (for image2audio generation) is achieved as below for each unmasking iteration, which operates in the prediction space output by the transformer.

$$\hat{p}_a = s \cdot p_a(\tilde{a}_i, v_i) + (1 - s) \cdot p_a(\tilde{a}_i, Mv_i) \tag{2}$$

where the $\tilde{a}_i$ means the input audio tokens in the current iteration, $v_i$ is the guided image token embedding without being masked, $M$ is a full mask leading to unconditional input, and hyperparameter $s$ is the CFG guidance factor to control the balance. As illustrated in Fig 3, there are actually 2 forward passes to transformer every iteration, where the two inputs are audio tokens concatenated with guided visual tokens and fully masked visual tokens.

Table 1: Accuracy on the VGGSound test set, where FD is the main metric. **T** means transformer. **CFG** means classifier-free guidance. ∗ means the results are reproduced with the official code and checkpoints, and the same visual data (video or the middle frame) is used for generation for all methods in the table. †: VAE 83.7M, UNet: 859.5M, Conditional part 0.4M, CAVP 113.4M. ††: VQGAN 72.1M, SpecVQGAN 54.5M, transformer: 424.4M.

| Method | Arch | #PARAM | Training | CFG Training | FD↓ | FAD↓ | IS↑ | CS↑ |
|---|---|---|---|---|---|---|---|---|
| Im2Wav* (Sheffer & Adi, 2023) | 2 autoregressive **T** | 361M | 1-stage | ✓ | 25.06 | 5.39 | 7.33 | **11.05** |
| DiffFoley* (Luo et al., 2023) | U-Net Diffusion | 1057M † | 2-stage | ✓ | 23.74 | 7.08 | 11.80 | 9.74 |
| **Ours w/o CFG** | Bi-directional **T** | 551M †† | 1-stage | ✗ | **16.14** | **1.79** | 8.30 | 10.10 |
| **Ours w/ CFG** | Bi-directional **T** | 551M †† | 1-stage | ✗ | **14.79** | **1.29** | **12.06** | 10.23 |

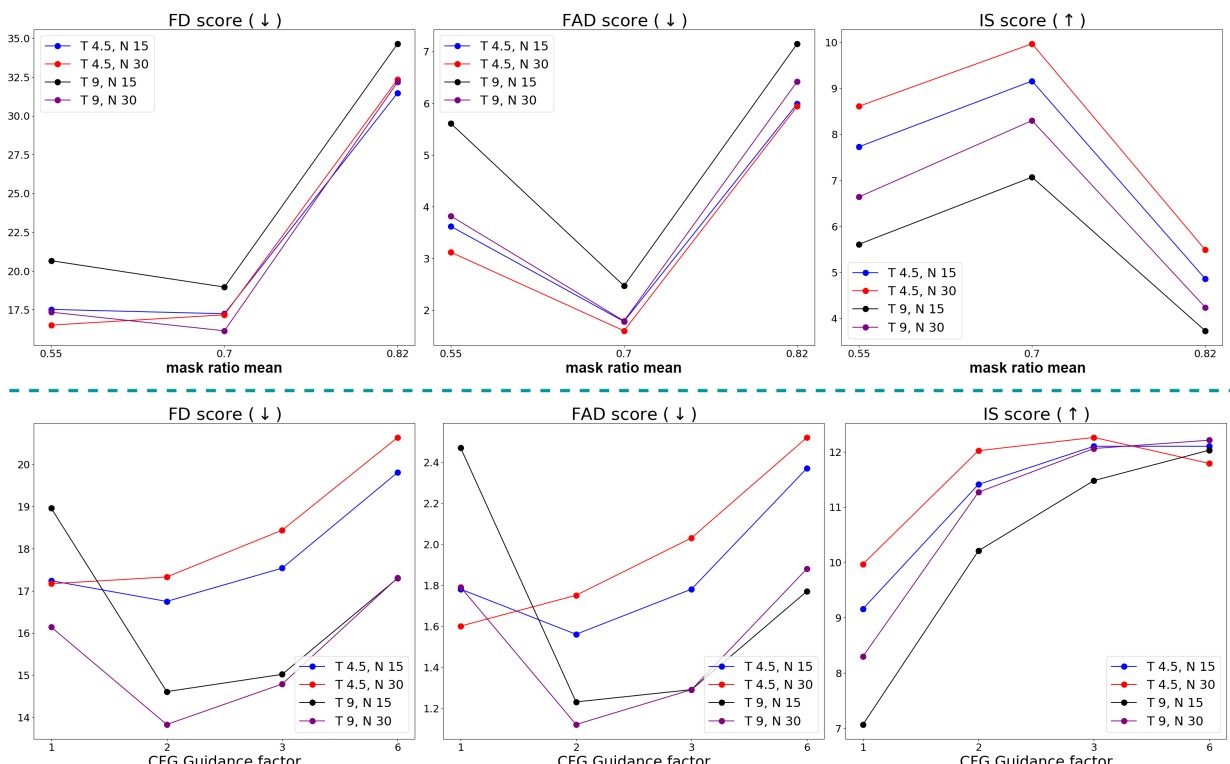

Figure 4: **(Top)** Ablation study on the mask ratio during training, no classifier-free guidance training or inference are used. $T$ and $N$ in the figure denote the temperature and sampling iteration number during inference. **(Bottom)** Ablation study on classifier-free Guidance (CFG) guidance factor, where guidance factor of 1 means not applying CFG.

# 4 Experiments

In this paper, we focus on ***image2audio generation*** task, while we will also show the results on ***audio2image generation*** and ***audio-image co-generation***. We will exclude methods using huge models, such as LLM and composable diffusion models. We only compare to the mostly related methods on image2audio generation, which is based on either autoregressive model (Sheffer & Adi, 2023) or latent diffusion model (Luo et al., 2023). The used image VQGAN which is pretrained on ImageNet, which generalizes badly on the dataset we use. Thus, for audio2image generation task, we will not directly compare to any specialized image generation model, while we do compare our method to a related audio2image generation method (Sung-Bin et al., 2023). More comparison results can be found in the appendix.

## 4.1 Experimental Details

**Dataset** We train and evaluate our method on the VGGSound (Chen et al., 2020) dataset, which is a widely used audio-visual dataset. It is a sound event curated dataset which has 309 sound events. The original VGGSound dataset contains video from YouTube where 183,727 videos in the train set and 15,446 videos for test set. Some of the videos are not existing anymore, for the VGGSound we use, there are 182,563 and 15,290 videos for train/test set respectively.

Table 2: Ablation study on Classifier-free Guidance (CFG) whether there are extra training for CFG and using CFG during inference or not. $\mu$ means the mean of the truncated Gaussian distribution used during training. **Hyper-para** means hyperparameters ($N$ for sampling iterations number and $T$ for temperature.) during inference. In all experiments, the CFG guidance factor is set to 3.

| $\mu$ | Hyper-para | CFG Training | CFG Inference | FD↓ | FAD↓ | IS↑ |
|---|---|---|---|---|---|---|
| 0.55 | $T$=4.5, $N$=15 | ✗ | ✗ | 17.52 | 3.62 | 7.73 |
| 0.55 | $T$=4.5, $N$=15 | ✗ | ✓ | 23.12 | 4.63 | 8.46 |
| 0.55 | $T$=4.5, $N$=15 | ✓ | ✓ | 20.83 | 2.00 | 8.82 |
| 0.55 | $T$=4.5, $N$=30 | ✗ | ✗ | 16.51 | 3.12 | 8.61 |
| 0.55 | $T$=4.5, $N$=30 | ✗ | ✓ | 23.78 | 4.71 | 8.72 |
| 0.55 | $T$=4.5, $N$=30 | ✓ | ✓ | 20.81 | 1.91 | 9.16 |
| 0.7 | $T$=4.5, $N$=15 | ✗ | ✗ | 17.24 | 1.78 | 9.16 |
| 0.7 | $T$=4.5, $N$=15 | ✗ | ✓ | 17.54 | 1.78 | 12.10 |
| 0.7 | $T$=4.5, $N$=30 | ✗ | ✗ | 17.17 | 1.60 | 9.97 |
| 0.7 | $T$=4.5, $N$=30 | ✗ | ✓ | 18.44 | 2.03 | 12.26 |

As mentioned in Sec. 3.1, we preprocess the audio to mel spectrogram, and uniformly sample 10 frames per video (all videos are 10s). During training, we randomly sample one frame for using, and during evaluation, we will always use the middle frame. The duration of the audio for training and generation is 10s.

**Tokenizer** For the visual part, we directly take the ImageNet pretrained VQGAN from (Esser et al., 2021), which has codebook of size 1024 and hidden dimension 256, the downsample ratio is 8. For audio part, we use the SpecVQGAN (Iashin & Rahtu, 2021) and a vocoder (HiFiGAN (Kong et al., 2020)) which are pretrained on AudioSet (Gemmeke et al., 2017). SpecVQGAN configuration is following the original paper (Iashin & Rahtu, 2021), which has a codebook of size 1024, each of which is represented by a 256-dim embedding. Two VQGANs are fixed during training.

**Transformer** We follow the ViT (Dosovitskiy et al., 2020) to build the transformer model, specifically we use the large size ViT in MAE (He et al., 2022). As for two embedding layers where the hidden dimension is 1024, their codebook size is $C + 1$ where $C$ is the corresponding codebook size of VQGAN and the extra one is for mask token. We train the transformer model for 1000 epochs on V100 (16G memory) nodes.

**Hyperparameters** The hyperparameters are the distribution mean $\mu$ for mask ratio sampling during training, the temperature $T$ and iterations $N$ for iterative inference, and $s$ for classifier-free guidance. If not specifically mentioned, $\mu, T, N, s$ are set to 0.7, 9, 30 and 3 respectively.

## 4.2 Quantitative Results

### 4.2.1 Image2Audio Generation

**Main Results** We compare our method to two current state-of-the-art image2audio method Im2Wav (Sheffer & Adi, 2023), and also a video2audio generation method DiffFoley (Luo et al., 2023). Since the used VGGSound dataset across methods may be different, as some videos may be missing, for fair comparison, we report the results of Im2Wav and DiffFoley by using their official code and checkpoint on the VGGSound dataset at our hand. DiffFoley directly takes the video as input, and for Im2Wav we use the middle frame image as its input. Our results are average over two random runnings.

The image2audio generation results are shown in Tab. 1. We list the architecture information. We adopt two widely used metrics in the audio communities for comparison, FAD (Frechet Audio Distance) and FD (Frechet Distance) which measure similarity between generated samples and target samples (VGGSound test set), following AudioLDM (Liu et al., 2023) FD will be the main metric. We also report IS score (Inception Score)[1] which measures both quality and diversity. We use the official evaluation repository of AudioLDM (Liu et al., 2023) to compute these metrics. In Tab. 1, our method with or without classifier-free guidance achieves the best performance on all three metrics, except that without CFG our method is worse on IS compared to DiffFoley, which is a video2audio generation method. Note our method does not demand either 2-stage training or larger parameters as DiffFoley, where a contrastive audio-video pretraining is needed and then an audio LDM is trained conditioned on video features, and our inference pipeline is

---

[1]We directly use this repo for computing, though it is not strictly correct as the class numbers of VGGSound and their training dataset are different, we still report it for reference.

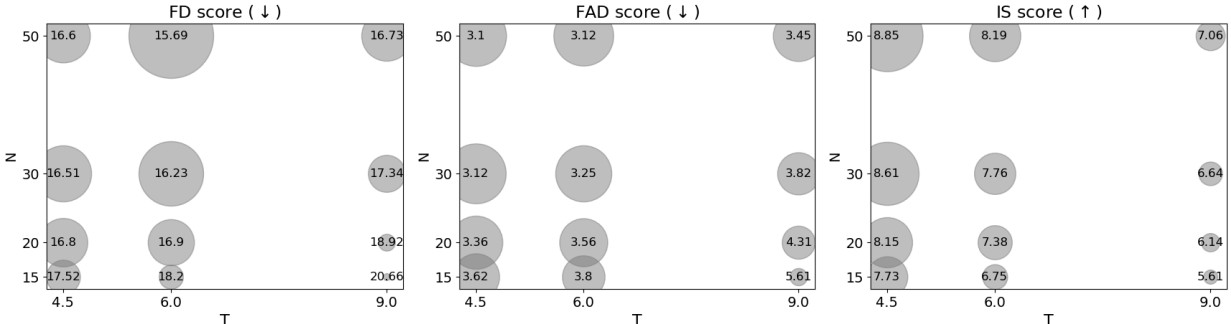

Figure 5: Ablation study on different temperature ($T$) and number of sampling iterations ($N$) during inference. The x and y axes denote $T$ and $N$ respectively. Larger size of the circle indicates better performance for the specific metric. The model here is trained with mask ratio sampled from the truncated Gaussian of mean 0.55, and no CFG training or inference is deployed.

expected to be faster than Im2Wav where there are 2 cascaded autoregressive transformers (one for low quality and one for high quality). The table also shows that CFG improves the performance a lot, and it is totally off-the-shelf, whereas DiffFoley needs extra training for CFG (and it also has extra training for classifier guidance.) Additionally, we also measure the CS score (similar to Clip Score) following Im2Wav (Sheffer & Adi, 2023) to measure the image and audio alignment. CS score is the average cosine feature similarity (multiplied by scaling factor 100) between generated audios and images which are used as condition, where the image feature is from CLIP (Radford et al., 2021) and audio feature is from Wav2CLIP (Wu et al., 2022) which has an audio encoder aligned with the CLIP image encoder. Simply using concatenating for audio-visual intersection, our method achieves a bit worse performance compared to Im2Wav on CS while still surpasses DiffFoley.

**Ablation on $\mu$** In Fig. 4 (**Top**), we show how different truncated Gaussian mean $\mu$, where mask ratio is sampled during training, influences the performance. It indicates that slighter larger $\mu$ will have better performance, while too larger will worsen the performance significantly.

**Ablation study on CFG** In Fig. 4 (**Bottom**), we show the results with varying CFG guidance factor $s$ under different inference hyperparameters. For IS score, larger guidance factor usually has better performance, while for FAD and FD smaller guidance factor needs to be set. The results show that with CFG the performance of all metrics are improved.

We also show the results with different strategies for CFG during training and inference in Tab. 2. We can draw 3 conclusions from the table. First, CFG mainly improve the IS scores, and may decrease other two metrics. Second, with larger truncated Gaussian mean $\mu$ for mask ratio sampling, there is no need to conduct extra training for CFG, where CFG could lead to decent performance on all metrics. Third, with small $\mu$, extra training for CFG could help to avoid lowering the FAD score. Overall, training with larger $\mu$ and inference with CFG can achieve good performance.

**Ablation on inference hyperparameters** We show the results with different inference hyperparameters $T, N$ (without CFG for clear ablation study) in Fig. 5. $N$ is the iteration number, and with $T$ there is more diversity introduced into the generation process (see Code. 1 in appendix). The figure indicates that slightly smaller $T$ and more iteration number $N$ can lead to better performance. Note in the main results, we choose $N = 30$ to balance the inference speed and performance.

**Discussion about Related Works** The most relevant existing work on visual-to-audio generation is FoleyGen (Mei et al., 2023), which is an autoregressive model (next token prediction) which is expected to be much slower during inference than us. And it gets worse performance (FAD 1.65) than us (FAD 1.29).

### 4.2.2 Audio2Image Generation and Co-Generation

We also show the results on audio2image generation and audio-image co-generation. We would like to emphasize again that it is not the main focus in this paper, and the used VQGAN generalizes badly on VGGSound dataset. In all experiments for image generation, we do not use CFG. Generated samples could be found in this anonymous link.

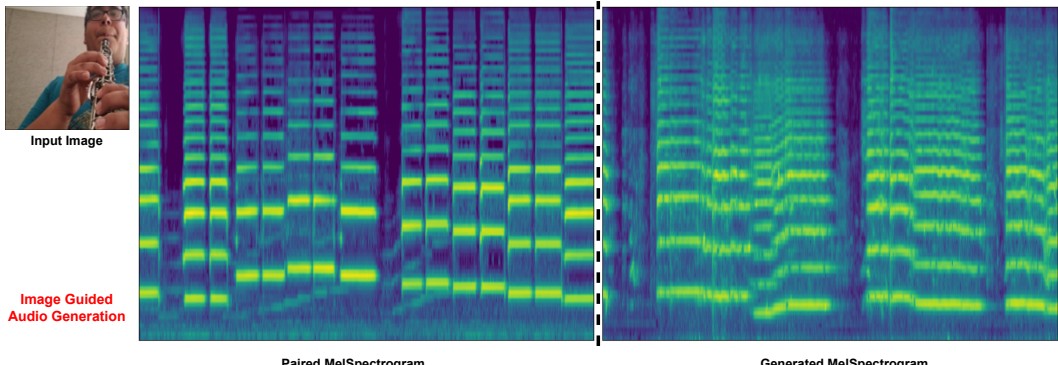

Figure 6: Visualization of mel spectrogram in image2audio generation. We show the input image, its corresponding groundtruth audio mel spectrogram and the generated mel spectrogram.

Table 3: Results on audio2image generation. We follow the setting of Sound2Scene with only using the images from **50 classes** in VGGSound.

| Method | Arch | FID↓ | IS↑ |
|---|---|---|---|
| Sound2Scene* (Sung-Bin et al., 2023) | Conv GAN | 62.95 | 19.15 |
| **Ours,** $N = 15, T = 4.5$ | Unified transformer | **40.69** | **20.62** |
| **Ours,** $N = 30, T = 4.5$ | Unified transformer | **42.02** | 19.14 |
| **Ours,** $N = 15, T = 9$ | Unified transformer | **36.22** | **24.25** |
| **Ours,** $N = 30, T = 9$ | Unified transformer | **35.63** | **23.47** |

Table 4: Results of Audio-Image Co-Generation. There is no classifier-free guidance deployed.

| | Image | | Audio | | |
|---|---|---|---|---|---|
| Method | FID↓ | IS↑ | FD↓ | FAD↓ | IS↑ |
| **Ours,** $N = 15, T = 4.5$ | 23.25 | 18.85 | 19.42 | 2.87 | 8.01 |
| **Ours,** $N = 30, T = 4.5$ | 24.21 | 17.54 | 18.74 | 2.29 | 9.46 |
| **Ours,** $N = 15, T = 9$ | 25.72 | 21.42 | 24.60 | 5.00 | 5.68 |
| **Ours,** $N = 30, T = 9$ | 20.86 | 20.78 | 20.42 | 3.54 | 6.99 |

For audio2image generation, we follow Sound2Scene (Sung-Bin et al., 2023), that we only use 50 specific classes in VGGSound. In Tab. 3, we show our method is better than Sound2Scene on both metrics, where FID measures fidelity by the similarity between generated images and images from VGGSound test set, and IS measures both fidelity and diversity. IS is computed with the model from Sound2Scene which is trained with the corresponding 50 classes. Sound2Scene takes a set of carefully picked frames to train the model, while our method does not demand it. For image-audio co-generation, we show the results in Tab. 4. Unlike Audio2Image generation where we only use 50 classes, here we generate 15290 samples which is just the same amount of test set and should cover all sound events.

**Discussion about Related Works** The most relevant existing work on joint audio-visual generation is MM-Diffusion (Ruan et al., 2023), which is only trained on Landscape (2.7 hours video/audio) and AIST++ (5.2 hours audio/video) which are much smaller than the VGGSound (around 507 hours audio/video) we use. Thus, we think it is not fair to compare with MM-Diffusion. In Tab. 4, we use 15290 videos from VGGSound test set to compute all those metrics, since our model is trained on VGGSound and it should generate similar samples. Those scores for MM-Diffusion are expected to be quite lower, as they are not trained on VGGSound which is larger and more complex than Landscape and AIST++. Since our submission is quite different from MM-Diffusion on both settings/tasks and trained data, we do not and could not compare to MM-Diffusion.

## 4.3 Qualitative Results

**Reconstruction visualization** We show the image and audio mel spectrogram reconstruction during training in Fig. 8 in the appendix. It indicates that the training objective is enough to reconstruct both modalities. The image reconstruction quality is bad, we ascribe it to the poor generalization of the utilized image VQGAN.

**Visualization of mel spectrogram** We visualize the audio mel spectrogram for image2audio generation in Fig. 6 in appendix. We show the input image, its corresponding groundtruth audio mel spectrogram and the generated mel spectrogram. It shows that the generated mel spectrogram with image guided has similar pattern as the groundtruth, indicates the visual-audio alignment is realized.

**Reconstruction quality of image VQGAN** We show the image reconstructed directly by VQGAN in Fig. 7 in the appendix, fine-grained detail information will be missing in someplace such as human face/finger and part of the instrument. We also show the direct reconstruction results with different pretrained VQGANs in Fig. 10 in the appendix,

where there are huge VQGAN models with 8192 codebook size and lower downsampled rate. The VQGAN used in the paper is the last one. All those model will lose information after reconstruction, while larger model indeed has better reconstruction quality. The phenomenon that pretrained VQGAN does not generalize well to images beyond trained data, is also observed by another work (Bai et al., 2024). The reconstruction quality on the VGGSound of utilized image VQGAN is bad. It will limit the image generation performance, as indicated by MAGVIT-v2 (Yu et al., 2024). We posit that with VQGAN of poor generalization, the error may be accumulated during inference leading to even worse image generation results. While the focus in this paper is image2audio generation, and the image VQGAN is already good enough to achieve desired image2audio generation results, we do not use the stronger image VQGAN to reduce the computation burden.

**Visualization of generated image** To give a visual illustration, we show the generated images from Sound2Scene (Sung-Bin et al., 2023), our method, and also some VQGAN reconstructed images in Fig. 9 in the appendix. The pretrained image VQGAN upperbounds the generation quality.

**Inpainting** After training, the model could also be deployed for inpainting task, since inpainting is a special case of generation. We adopt a naive way to achieve inpainting, for image-guided audio inpainting, we mask the input audio mel spectrogram and the audio tokens of corresponding area. We put inpainting/outpainting examples in this anonymous link.

# 5 Conclusion

In this paper, we investigate audio-visual generation. Instead of designing a giant model, we show a simple mask generative transformer could achieve good performance on image2audio generation. After training, the model could conduct image2audio, audo2image and image-audio co-generation, as well as inpainting task. Classifier-free guidance can be deployed off-the-shelf to further improve performance.

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

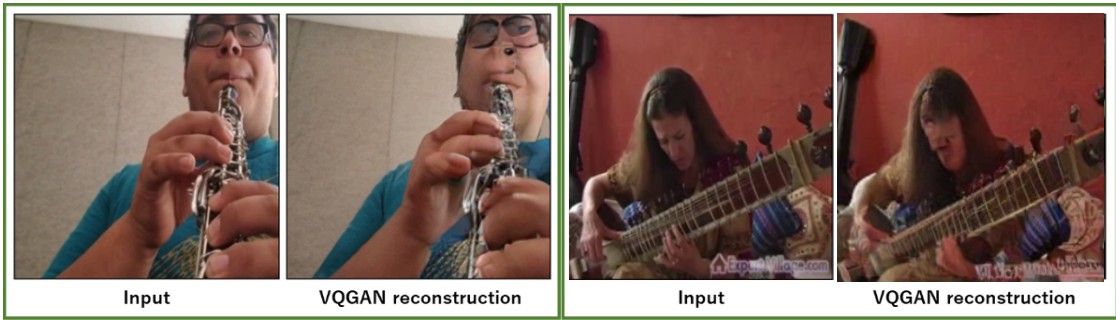

Figure 7: Reconstrution of image VQGAN, where the VQGAN directly takes in the image and output the reconstructed one.

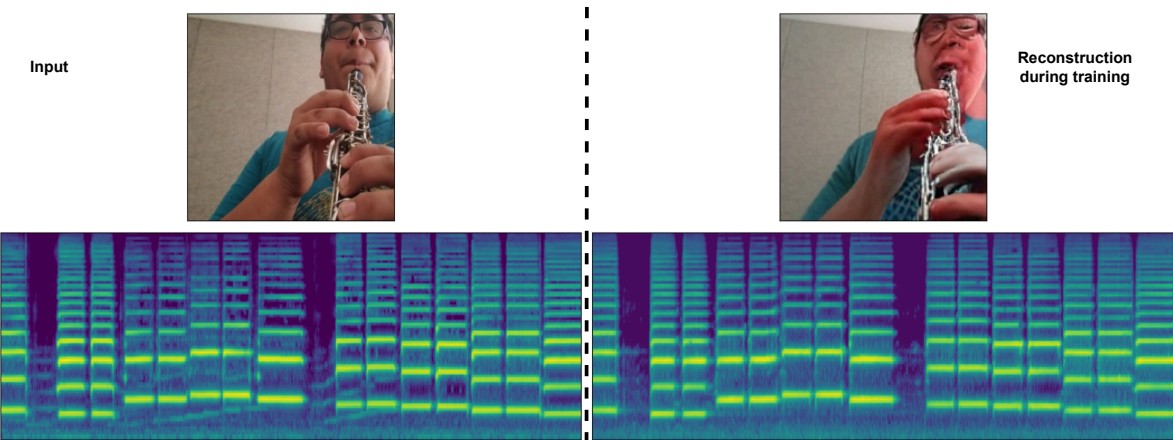

Figure 8: Image and audio mel spectrogram reconstruction during training.

## A  Appendix

### A.1  Discussion

**Limitations** The obvious limitation in current method is the poor image generation quality, we ascribe it to the poor generalization/reconstruction ability of ImageNet pretrained VQGAN on the VGGSound dataset. Another limitation is that the current pipeline can not deal with video data, which is a future direction to the current method. Additionally, current method lacks of the fine-grained visual controlled ability for audio generation, which is not investigated either in existing methods. Efforts need to be put in this direction to create algorithms of high real-world value.

Another limitation comes from the VGGSound dataset, there are videos which are totally black or black in certain time period. The consequence of training with this kind of video is that sometimes the model will generate black images

**Potential Issues** In this submission, we are using the VGGSound dataset, which contains videos originally from YouTube. As the official website of VGGSound mentioned, it is not possible to notify data subjects that content may have been used. While the University of Oxford (the dataset collector) has a data protection exemption on providing the notice directly to data subjects. This exemption is in Article 14(5)(b) of the UK GDPR. Though, the people contained in the video may ask for removing the video in this public dataset. Overall, there is no copyright issue currently.

Since the audio in the dataset may contain some music fragments, the AI model has possibility (while quite lower) to directly copy the style or pattern of those musics. Besides, AI models trained on those music data can be considered derivative works. There is ongoing debate and legal uncertainty regarding whether using a dataset to train an AI model constitutes the creation of a derivative work, but the risk of infringement claims exists. While judging by the video duration (10s), it is of low possibility to contain a full music fragment, the risk of derivative issue is limited.

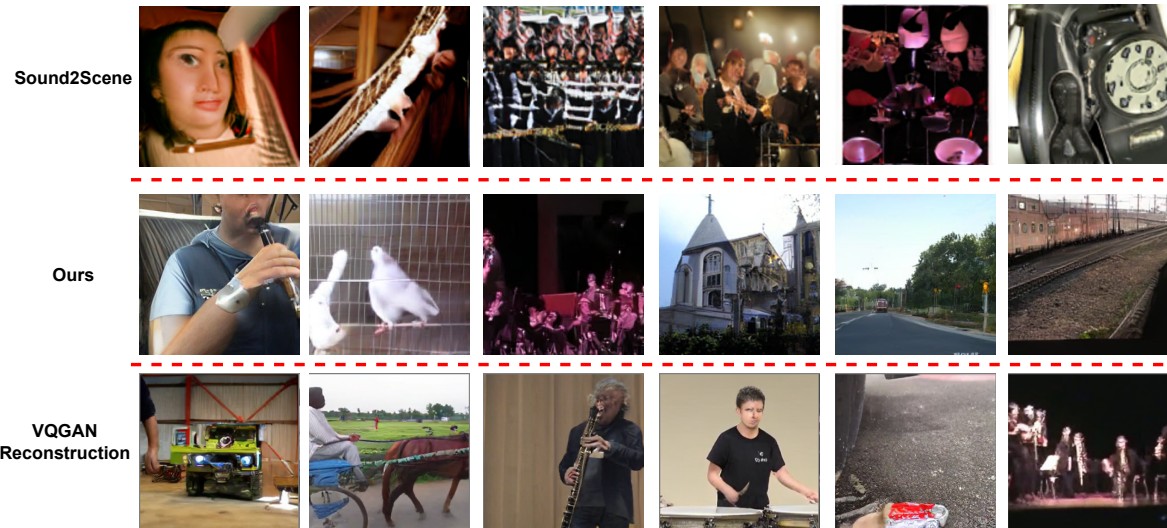

Figure 9: (**Top**) Generated images from Sound2Scene. (**Middle**) Generated images from our methods. (**Bottom**) Reconstructed image from image VQGAN.

Generally speaking, the potential for AI-generated audio to be used for fraudulent purposes is a growing concern, with capabilities including impersonation, enhanced phishing scams, and the creation of convincing misinformation. However, in this paper, most of the training data does not have high quality human speech and the capacity of the current model is far away from synthesizing realistic level speech. This possibility of this risk is quite low.

## A.2   Discussion and Additional Results

**Training time and GPU memory comparison** According to Diff-Foley paper, the LDM training takes 80 hours on 8 A100 GPUs with a total batch size of 1760. While the training of our model takes  29 hours on 128 V100 (16 GB VRAM) with a total batch size of 512. Diff-Foley can only address video2audio generation task, while our method can achieve multiple tasks. As we do not have long-term usable GPUs, we do not run ablation study on the amount of training epochs.

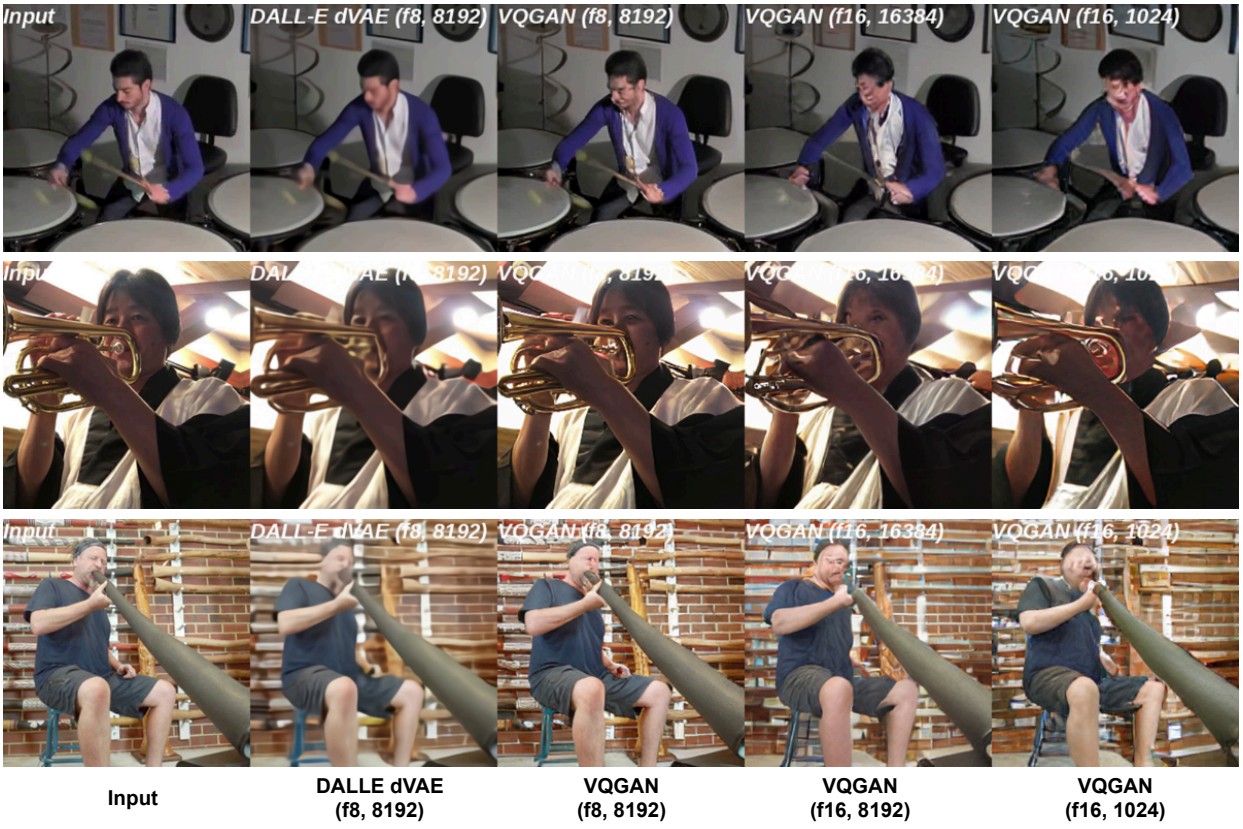

Figure 10: Reconstruction on VGGSound of different discrete models. In the paper, we are using the rightmost one. $f8$ and $f16$ means downsample ratio in the encoder, and the $8192$ and $1024$ means the size of the codebook.

