# OpenReview forum: "Visual Echoes: A Simple Unified Transformer for Audio-Visual Generation"
_TMLR — Rejected by TMLR_

### Review · Reviewer_ixcD · 2024-11-22

**Summary Of Contributions:**

The authors provide a simple and small (~0.5B parameters) audio-image joint modality model trained using a generative transformer. Though the results largely focus on audio2img application, the model architecture and training allows for flexible conditional inference. Moreover, it allows for application of off-the-shelf classifier-free guidance. The reported results of audio2img are competitive or better than models of similar size.

**Audience:**

Yes

**Broader Impact Concerns:**

Authors have sufficiently addressed the concerns.

**Claims And Evidence:**

Yes

**Requested Changes:**

The presented work is a good technical study. I only have one major comment — given the claim in appendix “can achieve excellent results on image2audio generation” and conclusion “The proposed method could be adopted as a simple while strong baseline for audio-visual generation research in the future.”, I would have expected some comparison numbers with stronger baselines. However, authors refrain to do so: “We will exclude methods using huge models, such as LLM and composable diffusion models.” for img2audio task and compare against only one audio2img model. It will be good to either provide comparison points with bigger models or models used in related tasks to quantify the gap, or tone down these statements.

### Minor
Technical questions and/or suggestions, in no-particular order:
- Which positional embedding was used?
- S3.4, “Conditional generation”, typo: “where in every iteration a certain ratio of the transformer predictions token will be updated to the masked input with remaining tokens kept masked” -> will be updated to unmasked input
- Figure 3 is unclear to me and I would highly recommend re-doing this. “CFG” can also be expanded in the caption.
- S4, “Inpainting”, typo: “ After training, the model also be deployed for inpainting task” -> the model will also be deployed
- Appendix, Fig 10, Caption typo: “Bottle” -> bottom

**Strengths And Weaknesses:**

### Strengths
- The problem is well-formulated and the past literature is summarized well.
- The solution, though simple, seems competitive with more complex methods.
- Ablations for hyper parameters and qualitative results are provided.

### Weaknesses
- Adding more comparisons to baselines can highlight the contributions better.

---

> ### Author Response · Authors · 2024-11-25
>
> We thank the reviewer for the valuable comments. We just upload the revised version, with edited parts marked with blue color, where the typos are fixed and overclaimed sentence in the conclusion is removed. We also provide a new figure to illustrate the Classifier-free Guidance.
>
> As for positional embedding, we use the learnable positional embedding, we added this narration in the revised version.
>
> As for more comparisons to baselines, since the other baseline methods are with different architecture, the major comparisons in the submission covered enough quantitative results (quality, model size, running time) and a few qualitative results (in the pdf and also in the anonymous link). We think those results could prove the effectiveness of the method. We add one sentence at the beginning of the experimental section saying more comparison results can be found in the appendix.

---

### Review · Reviewer_Uqdx · 2024-12-02

**Summary Of Contributions:**

This paper present a simple unifed transformer for joint audio and visual generation. The authors present a simple and  lightweghted generative transformer, which is trained via mask denoising manner. The generation is conducted in the space of the discrete audio and visual vector-quantized GAN space. Further, the classifer-free guidance could be deployed off-the-shelf to achieve better performance. Moreover, the proposed transformer model is modality symmetrical, so it could be directly deployed for audio2image generation and co-generation. Empirically the authors show that the proposed method surpasses recent image2audio generation methods.

**Audience:**

Yes

**Claims And Evidence:**

Yes

**Requested Changes:**

- Please use citet and citep properly in your papers
- Please discuss and compare with diffusion based approaches
- please clarify why the reconstruction images are so bad in your appendix.

**Strengths And Weaknesses:**

**Strength**
- The paper is well written and the method is simple and easy to understand.
- Compared with previous latent diffusion model based approaches, the masked denoising approach to train the transformers is much simpler and can easily connect two different modalities.
- The authors also proposed iterative masking and classifer-free guidance under this setting to achieve better inference performance


**Weakness**
- It is not clear how much advantage of the proposed method over diffusion based methods. It would be good to have a discussion
- From the appendix, we can see the reconstructed images from VQ-GAN is quite bad; The authors should have a discussion and investigation of whether it is because of the VQ-GAN training, or some other reasons.

---

> ### Author Response · Authors · 2024-12-05
>
> We thank the reviewer for the valuable comments. We put our response below.
>
> - It is not clear how much advantage of the proposed method over diffusion based methods. It would be good to have a discussion
>
> At the end of Sec. 3.3, we discuss the relation of our method to diffusion model, and our method could be regarded as a case of discrete diffusion. And in the experimental section, we compare our method to DiffFoley which is a diffusion based method. Note that DiffFoley is the most relevant work in visual-to-audio generation, while most other works focus on text conditional audio generation. The experimental results in Tab.1 show that our method has much better audio generation quality with less parameters. Another contribution of our method, is proving that besides diffusion model, masking modeling could also achieve audio-visual generation.
>
> - Please clarify why the reconstruction images are so bad in your appendix.
>
> We would like to emphasize that we directly use the VQ-GAN pretrained on imagenet, as stated at the starting of Sec. 3.2. We do not finetune the VQ-GAN since it is non-trivial. The phenomenon that imagenet pretrained VQ-GAN generalizes bad on images beyond imagenet dataset is also observed in another work [1]. We add this discussion in Sec. 4.3 with red color.
>
> [1]. Yutong Bai, Xinyang Geng, Karttikeya Mangalam, Amir Bar, Alan L Yuille, Trevor Darrell, Jitendra Malik, and
> Alexei A Efros. Sequential modeling enables scalable learning for large vision models. In Proceedings of the
> IEEE/CVF Conference on Computer Vision and Pattern Recognition, pp. 22861–22872, 2024.
>
> - Please use citet and citep properly in your papers
>
> Thanks for the advice, we fix the issue in the newly uploaded version.

---

### Review · Reviewer_UGZm · 2025-02-28

**Summary Of Contributions:**

This paper proposes a lightweight generative Transformer model for image-to-audio generation. The model leverages masked denoising training, iterative unmasking inference, and classifier-free guidance (CFG) to enhance the generation quality. Additionally, the proposed approach supports audio-to-image generation and audio-visual co-generation. Experimental results demonstrate that the proposed method achieves better generation performance compared to two existing baselines: Im2Wav and DiffFoley.

**Audience:**

Yes

**Broader Impact Concerns:**

No ethical concerns.

**Claims And Evidence:**

No

**Requested Changes:**

1.Clearly define the motivation and significance of image-to-audio generation, including real-world applications where this task is useful.
2. Include more recent works related to image-to-audio generation, and explicitly discuss how this work differs from them.
3. Expand the dataset evaluation beyond VGGSound, demonstrating that the proposed framework remains effective across different datasets.
4. Add more baselines to benchmark against newer image-to-audio generation methods, beyond just Im2Wav and DiffFoley.

**Strengths And Weaknesses:**

Strengths:
-The paper is easy to understand, and the figures clearly illustrating the implementation details.
-The authors provide a detailed description of the method part, making it easier for readers to follow.

Weaknesses:
1.Unclear Motivation
The motivation behind image-to-audio generation is not well-explained. While video-to-audio generation has a clear practical application (e.g., automated sound dubbing for videos), it is unclear what real-world applications exist for generating audio from a single static image. The authors should clarify why this task is meaningful and provide concrete examples of its potential applications.

2.Lack of Novelty in the Proposed Method
The proposed method appears to directly transfer form exsisting method. The discrete tokenization is based on existing models, and masked denoising training is a standard technique. Neither the training structure nor the optimization loss function appears to be uniquely designed for this specific task. The authors should clearly highlight any methodological innovations beyond its direct application to image-to-audio generation.

3.Limited Baseline Comparisons
The experimental evaluation only includes two baselines (Im2Wav and DiffFoley) from 2023. Given the rapid progress in multi-modal generative models, more recent methods from 2024 or 2025 should be included for a more comprehensive comparison. Please see these reference: [1][2][3][4], etc.

4.Limited Dataset Evaluation
The model is only evaluated on VGGSound, which raises concerns about its generalizability to other datasets.
To verify the robustness of the proposed approach, the authors should test the model on additional datasets.

5.Unclear Data Sampling Strategy
Some data process details are unclear. How many audio-image pairs were used from VGGSound?
Was each video in VGGSound paired with only one audio-image pair sample, or were multiple pair for one video sampled?
If only one frame is selected from the 10 video frames, does choosing a different frame significantly impact the model’s performance?

[1]"Seeing and Hearing: Open-domain Visual-Audio Generation with Diffusion Latent Aligners (CVPR 2024)".
[2]“MeLFusion: Synthesizing Music from Image and Language Cues using Diffusion Models,” CVPR 2024​
[3]'Images that Sound: Composing Images and Sounds on a Single Canvas. NeurIPS 2024
[4]Tell What You Hear From What You See - Video to Audio Generation Through Text. NeurIPS 2024

---

> ### Author Response · Authors · 2025-02-28
>
> Thank the reviewer for the valuable comments. We put our response below.
>
> 1. **Unclear Motivation and Potential Application Cases** \
> The submission is a preliminary work towards future audio-video generation, by proving that the masked generative transformer works well on audio-image generation tasks. The potential application cases of audio-image generation mainly lie in entertainment area. There already exists some business applications such as artistic creation which can be used in multisensory art showing, and also sound design where the generated audio could be used as the starting samples for later more fine-grained designing. Also, it could also be used in assistance service for visually impaired individuals.
>
> 2. **Lack of Novelty** \
> We acknowledge that our work does not introduce a plethora of novel modules. Our deliberate choice to keep the pipeline as simple as possible ensures easy extension to future downstream tasks, such as incorporating text conditions. More importantly, our main contribution lies in demonstrating that a simple masked generative transformer can perform effectively. Given that many large-scale models in visual generation share similar architectures—and that performance improvements often stem from high-quality data—we believe that a simple streamlined pipeline offers significant value in (especially) industrial research.
>
> 3. **Limited Baseline Comparisons** \
> We added more results from the recent methods as below. Note that MeLFusion [2] focuses on music and only reports results on music benchmark, and Images that Sound [3] is not a specific generation method which has bad performance compared to generation specific methods, we do not include these methods for comparison. For seeing-and-hearing (Image-to-Audio) [1] and VATT-Gemma [4] (Video-to-Audio), we directly copy their results for comparison, and only reports the FAD and FD which measure the audio quality. The results indicate that our method works quite well.
>
>
>     Method  | FD | FAD
>     -------------------|------------------|------------------
>     Im2Wav       |        25.06    |         5.39
>     DiffFoley       |       23.74     |         7.08
>     Seeing-and-Hearing  I2A  [1]   |     20.96    |         6.87
>     VATT-Gemma  [4]     |     -      |         2.35
>     Ours    |      **14.79**     |         **1.29**
>
>     Our main difference from the above methods is, instead of resorting to multi-stage training and heavy models, we only use a
>     simple masked generative transformer, which even has better audio generation quality. The method could be used as a strong
>     baseline in future research.
>
>
> 4. **Limited Dataset Evaluation** \
> Currently we only run experiments on vggsound dataset, which is one of the challenging audio dataset as it contains many noisy videos from youtube. We posit that our method will also work well on other dataset, since we do not have any cumbersome heavy module which makes it easy to generalize to other benchmarks. We will add more experiments on AudioCaps in the future.
>
>
> 5. **Unclear Data Sampling Strategy** \
> As mentioned in Sec. 3.1 and Sec. 4.1, during training we sample 10 frames uniformly within a video and randomly pick one for training in every iteration which ensures robust paired image-audio training, during inference we will always use the middle frame of the video. We borrow this data pipeline from CAV-MAE [5].
>
>
> ### Reference
>
> [1] Seeing and Hearing: Open-domain Visual-Audio Generation with Diffusion Latent Aligners. CVPR 2024\
> [2] MeLFusion: Synthesizing Music from Image and Language Cues using Diffusion Models.  CVPR 2024​\
> [3] Images that Sound: Composing Images and Sounds on a Single Canvas. NeurIPS 2024 \
> [4] Tell What You Hear From What You See - Video to Audio Generation Through Text. NeurIPS 2024 \
> [5] Contrastive Audio-Visual Masked Autoencoder. ICLR 2023

---

### Decision · Action_Editor_8PsL · 2025-04-01

**Recommendation:** Reject

**Comment:**

First, I would like apologize to the authors on the delay of the reviewing process. There were some complications in the process by assigning and chasing the reviewers and get their inputs. Overall, the reviewers agree the paper proposes a simple framework for audio-visual generation with reasonable performance. One major concern from the reviewer is that the paper claims the proposed method surpasses recent image2audion generative models. However, from the current experimental results, the quality of the generated samples is sub-part and there is still room for improvement. The reviewers are not convinced that the proposed method is truly effective. For example, the evaluation is only conducted on the VGGSound dataset, which lacks evidence of the generalization ability. Furthermore, there are limited baseline comparisons (with Im2Wav and DiffFoley). The reviewer is asking for comparisons with more recent models. Although this is partially addressed in the rebuttal, a more comprehensive comparison by including more details such as reevaluating Seeing-and-Hearing and VATT-Gemma (directly taking results from the original papers are not fair), as well as including the IS and some visualization, is expected. In general, the authors are encouraged to scale up to training and model for better generations to demonstrate the effectiveness of different components.

Based on the above main concerns, I have to propose reject for this round of review. I encourage the authors to address the concerns and revise the paper for the next round submission.

**Audience:**

Researchers in generative models might find the work interesting. However, the current results are unlikely to make significant attention.

**Claims And Evidence:**

This paper proposes a light-weighted transformer based model for audio-visual generation, trained in the mask denoising manner. It claims to obtain excellent results on image2audio generation and demonstrates better performance compared to some baseline models. While the reviewers generally agree the simple and easily understanding of the proposed method, most of them show concerns on the empirical results. More details are listed below.

**Resubmission Of Major Revision:**

The authors may consider submitting a major revision at a later time.